# CRESCENDONET: A NEW DEEP CONVOLUTIONAL NEURAL NETWORK WITH ENSEMBLE BEHAVIOR

## ABSTRACT

We introduce a new deep convolutional neural network, CrescendoNet, by stacking simple building blocks without residual connections. Each Crescendo block contains independent convolution paths with increased depths. The numbers of convolution layers and parameters are only increased linearly in Crescendo blocks. In experiments, CrescendoNet with only 15 layers outperforms almost all networks without residual connections on benchmark datasets, CIFAR10, CIFAR100, and SVHN. Given sufficient amount of data as in SVHN dataset, CrescendoNet with 15 layers and 4.1M parameters can match the performance of DenseNet-BC with 250 layers and 15.3M parameters. CrescendoNet provides a new way to construct high performance deep convolutional neural networks with simple network architecture. Moreover, through investigating the behavior and performance of subnetworks in CrescendoNet, we note that the high performance of CrescendoNet may come from its implicit ensemble behavior. Furthermore, the independence between paths in CrescendoNet allows us to introduce a new path-wise training procedure, which can reduce the memory needed for training.

## 1 INTRODUCTION

Deep convolutional neural networks (CNNs) have significantly improved the performance of image classification (Krizhevsky et al., 2012; He et al., 2016a; Szegedy et al., 2015). However, training a CNN also becomes increasingly difficult with the network deepening. One of important research efforts to overcome this difficulty is to develop new neural network architectures (Huang et al., 2016a; Larsson et al., 2017).

Recently, residual network (He et al., 2016a) and its variants (Huang et al., 2017) have used residual connections among layers to train very deep CNN. The residual connections promote the feature reuse, help the gradient flow, and reduce the need for massive parameters. The ResNet (He et al., 2016a) and DenseNet (Huang et al., 2017) achieved state-of-the-art accuracy on benchmark datasets. Alternatively, FractalNet (Larsson et al., 2017) expanded the convolutional layers in a fractal form to generate deep CNNs. Without residual connections (He et al., 2016a) and manual deep supervision (Lee et al., 2014), FractalNet achieved high performance on image classification based on network structural design only.

Many studies tried to understand reasons behind the representation view of deep CNNs. Veit et al. (2016) showed that residual network can be seen as an ensemble of relatively shallow effective paths. However, Greff et al. (2017) argued that ensembles of shallow networks cannot explain the experimental results of lesioning, layer dropout, and layer reshuffling on ResNet. They proposed that residual connections have led to unrolled iterative estimation in ResNet. Meanwhile, Larsson et al. (2017) speculated that the high performance of FractalNet was due to the unrolled iterative estimation of features of the longest path using features of shorter paths. Although unrolled iterative estimation model can explain many experimental results, it is unclear how it helps improve the classification performance of ResNet and FractalNet. On the other hand, the ensemble model can explain the performance improvement easily.

In this work, we propose CrescendoNet, a new deep convolutional neural network with ensemble behavior. Same as other deep CNNs, CrescendoNet is created by stacking simple building blocks, called Crescendo blocks (Figure 1). Each Crescendo block comprises a set of independent feed-forward paths with increased number of convolution and batch-norm layers (Ioffe & Szegedy,

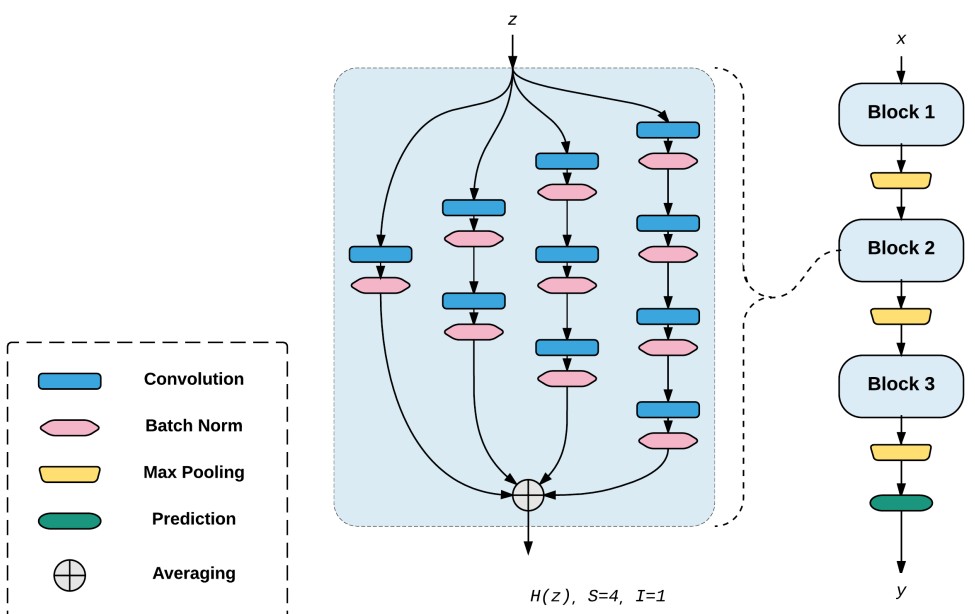

Figure 1: CrescendoNet architecture used in experiments, where $scale = 4$ and $interval = 1$.

2015a). We only use the identical size, $3 \times 3$, for all convolutional filters in the entire network. Despite its simplicity, CrescendoNet shows competitive performance on benchmark CIFAR10, CIFAR100, and SVHN datasets.

Similar to FractalNet, CrescendoNet does not include residual connections. The high performance of CrescendoNet also comes completely from its network structural design. Unlike the FractalNet, in which the numbers of convolutional layers and associated parameters are increased exponentially, the numbers of convolutional layers and parameters in Crescendo blocks are increased linearly.

CrescendoNet shows clear ensemble behavior (Section 3.4). In CrescendoNet, although the longer paths have better performances than those of shorter paths, the combination of different length paths have even better performance. A set of paths generally outperform its subsets. This is different from FractalNet, in which the longest path alone achieves the similar performance as the entire network does, far better than other paths do.

Furthermore, the independence between paths in CrescendoNet allows us to introduce a new path-wise training procedure, in which paths in each building block are trained independently and sequentially. The path-wise procedure can reduce the memory needed for training. Especially, we can reduce the amortized memory used for training CrescendoNet to about one fourth.

We summarize our contribution as follows:

- We propose the Crescendo block with linearly increased convolutional and batch-norm layers. The CrescendoNet generated by stacking Crescendo blocks further demonstrates that the high performance of deep CNNs can be achieved without explicit residual learning.

- Through our analysis and experiments, we discovered an emergent behavior which is significantly different from which of FractalNet. The entire CrescendoNet outperforms any subset of it can provide an insight of improving the model performance by increasing the number of paths by a pattern.

- We introduce a path-wise training approach for CrescendoNet, which can lower the memory requirements without significant loss of accuracy given sufficient data.

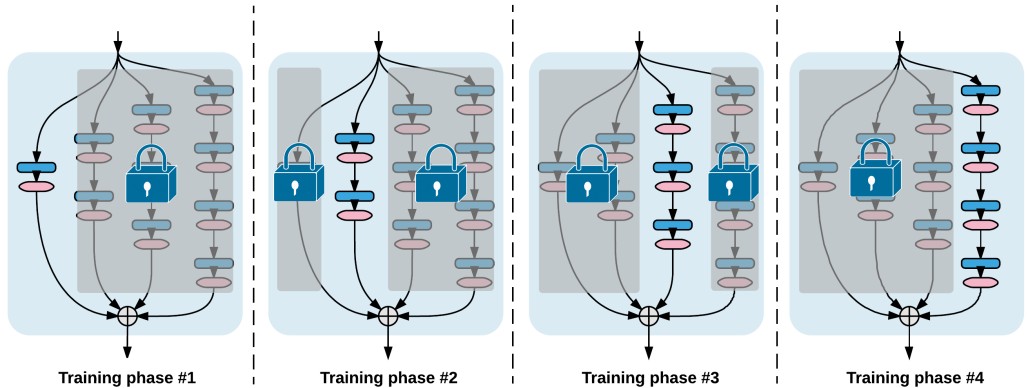

Figure 2: Path-wise training procedure.

# 2 CRESCENDONET

## 2.1 ARCHITECTURE DESIGN

**Crescendo Block** The Crescendo block is built by two layers, the convolution layer with the activation function and the following batch normalization layer (Ioffe & Szegedy, 2015b). The convolutional layers have the identical size, $3 \times 3$. The Conv-Activation-BatchNorm unit $f_1$, defined in the Eq.1 is the base branch of the Crescendo block. We use ReLU as the activation function to avoid the problem of vanishing gradients (Nair & Hinton, 2010).

$$f_1(z) = batchnorm(activation(conv(z))) \tag{1}$$

The variable $z$ denotes the input feature maps. We use two hyper-parameters, the scale $S$ and the interval $I$ to define the structure of the Crescendo block $H_S$. The interval $I$ specifies the depth difference between every two adjacent branches and the scale $S$ sets the number of branches per block. The structure of the $n^{th}$ branch is defined by the following equation:

$$f_n(z) = f_1^{nI}(z) \tag{2}$$

where the superscript $nI$ is the number of recursion time of the function $f_1$. The structure of Crescendo block $H_S$ can be obtained below:

$$H_S(z) = f_1(z) \oplus f_2(z) \oplus ... f_S(z) \tag{3}$$

where $\oplus$ denotes an element-wise averaging operation. Note that the feature maps from each path are averaged element-wise, leaving the width of the channel unchanged. A Crescendo block with $S = 4$ and $I = 1$ is shown in Figure 1.

The structure of Crescendo block is designed for exploiting more feature expressiveness. The different depths of parallel paths lead to different receptive fields and therefore generate features in different abstract levels. In addition, such an incremental and parallel form explicitly supports the ensemble effects, which shows excellent characteristics for efficient training and anytime classification. We will explain and demonstrate this in the following sections.

**CrescendoNet Architecture** The main body of CrescendoNet is composed of stacked Crescendo blocks with max-pooling layers between adjacent blocks (Figure 1). Following the main body, like most deep CNNs, we use two fully connected layers and a soft-max layer as the classifier. In all experiments, the two fully connected layers have 384 hidden units and 192 hidden units respectively. The overall structure of CrescendoNet is simple and we only need to tune the Crescendo block to modify the entire network.

## 2.2 PATH-WISE TRAINING

To reduce the memory consumption during training CrescendoNet, we propose a path-wise training procedure, leveraging the independent multi-path structure of our model. We denote stacked

Conv-BatchNorm layers in one Crescendo block as one path. We train each path individually, from the shortest to the longest repetitively. When we are training one path, we freeze the parameters of other paths. In other words, these frozen layers only provide learned features to support the training. Figure 2 illustrates the procedure of path-wise training within a CrescendoNet block containing four paths. There are two advantages of path-wise training. First, path-wise training procedure significantly reduces the memory requirements for convolutional layers, which constitutes the major memory cost for training CNNs. For example, the higher bound of the memory required for computation and storage of gradients using momentum stochastic gradient descent algorithms can be reduced to about 40% for a Crescendo block with 4 paths where $interval = 1$. Second, path-wise training works well with various optimizers and regularizations. Even dropout and drop-path can be applied to the model during the training.

### 2.3 REGULARIZATION

Dropout (Hinton et al., 2012) and drop-connect (Wan et al., 2013), which randomly set a selected subset of activations or weights to zero respectively, are effective regularization techniques for deep neural networks. Their variant, drop-path (Larsson et al., 2017), shows further performance improvement by dropping paths when training FractalNet.

We use both dropout and drop-path for regularizing the Crescendo block. We drop the branches in each block with a predefined probability. For example, given drop-path rate, $p = 0.3$, the expectation of the number of dropped branches is 1.2 for a Crescendo block with four branches. For the fully connected layers, we use L2 norm of their weights as an additional term to the loss.

## 3 EXPERIMENTS

### 3.1 DATASETS

We evaluate our models with three benchmark datasets: CIFAR10, CIFAR100 (Krizhevsky et al., 2014), and Street View House Numbers (SVHN) (Netzer et al., 2011). CIFAR10 and CIFAR100 each have 50,000 training images and 10,000 test images, belonging to 10 and 100 classes respectively. All the images are in RGB format with the size of $32 \times 32$-pixel. SVHN are color images, with the same size of $32 \times 32$-pixel, containing 604,388 and 26,032 images for training and testing respectively. Note that these digits are cropped from a series of numbers. Thus, there may be more than one digit in an image, but only the one in the center is used as the label. For data augmentation, we use a widely adopted scheme (Lin et al., 2013; Larsson et al., 2017; Huang et al., 2016a;b; Srivastava et al., 2015b; Springenberg et al., 2014; He et al., 2016a). We first pad images with 4 zero pixels on each side, then crop padded images to $32 \times 32$-pixel randomly and horizontally flipping with a 50% probability. We preprocess each image in all three datasets by subtracting off the mean and dividing the variance of the pixels.

### 3.2 TRAINING

We use Mini-batch gradient descent to train all our models. We implement our models using TensorFlow distributed computation framework (Abadi et al., 2016) and run them on NVidia P100 GPU. We also optimize our models by adaptive momentum estimation (Adam) optimization (Kingma & Ba, 2014) and Nesterov Momentum optimization (Nesterov, 1983) respectively. For Adam optimization, we set the learning rate hyper-parameter to 0.001 and let Adam adaptively tune the learning rate during the training. We choose the momentum decay hyper-parameter $\beta_1 = 0.9$ and $\beta_2 = 0.999$. And we set the smoothing term $\epsilon = 10^{-8}$. This configuration is the default setting for the AdamOptimizer class in TensorFlow. For Nesterov Momentum optimization, we set the hyper-parameter $momentum = 0.9$. We decay the learning rate from 0.1 to 0.01 after 512 epochs for CIFAR and from 0.05 to 0.005, then to 0.0005, after 42 epochs and 63 epochs respectively for SVHN. We use truncated normal distribution for parameter initialization. The standard deviation of hyper-parameters is 0.05 for convolutional weights and 0.04 for fully connected layer weights. For all datasets, we use the batch size of 128 on each training replica. For the whole net training, we run 700 epochs on CIFAR and 70 epochs on SVHN. For the path-wise training, we run 1400 epochs on CIFAR and 100 epochs on SVHN.

Using a CrescendoNet model with three blocks each contains four branches as illustrated in Figure 1, we investigate the following preliminary aspects: the model performance under different block widths, the ensemble effect, and the path-wise training performance. We study the Crescendo block with three different width configurations: equal width globally, equal width within the block, and increasing width. All the three configurations have the same fully connected layers. For the first one, we set the number of feature maps to 128 for all the convolutional layers. For the second, the numbers of feature maps are (128, 256, 512) for convolutional layers in each block. For the last, we gradually increase the feature maps for each branch in three blocks to (128, 256, 512) correspondingly. For example, the number of feature maps for the second and fourth branches in the second block is (192, 256) and (160, 192, 224, 256). The exact number of maps for each layer is defined by the following equation:

$$n_{maps} = n_{inmaps} + i_{layer}\frac{n_{outmaps} - n_{inmaps}}{n_{layers}} \tag{4}$$

where $n_{maps}$ denotes the number of feature maps for a layer, $n_{inmaps}$ and $n_{outmaps}$ are number of input and output maps respectively, $n_{layers}$ is the number of layers in the block, and $i_{layer}$ is the index of the layer in the branch, starting from 1.

To inspect the ensemble behavior of CrescendoNet, we compare the performance of models with and without drop-path technique and subnets composed by different combinations of branches in each block. For the simplicity, we denote the branch combination as a set $P$ containing the index of the branch. For example, $P = \{1, 3\}$ means the blocks in the subnet only contains the first and third branches. The same notation is used in Table 2 and Figure 3.

### 3.3 RESULTS OF THE WHOLE NET

Table 1 gives a comparison among CrescendoNet and other representative models on CIFAR and SVHN benchmark datasets. For five datasets, CrescendoNet with only 15 layers outperforms almost all networks without residual connections, plus original ResNet and ResNet with Stochastic Depth. For CIFAR10 and CIFAR100 without data augmentation, CrescendoNet also performs better than all the given models except DenseNet with bottleneck layers and compression (DenseNet-BC) with 250 layers. However, CrescendoNet's error rate 1.76% matches the 1.74% error rate of given DenseNet-BC, on SVHN dataset which has plentiful data for each class. Comparing with Fractal-Net, another outstanding model without residual connection, CrescendoNet has a simpler structure, fewer parameters, but higher accuracies.

The lower rows in Table 1 compare the performance of our model given different configuration. In three different widths, the performance simultaneously grows with the number of feature maps. In other words, there is no over-fitting when we increase the capacity of CrescendoNet in an appropriate scope. Thus, CrescendoNet demonstrates a potential to further improve its performance by scaling up. In addition, the drop-path technique shows its benefits to our models on all the datasets, just as it does to FractalNet.

Another interesting result from Table 1 is the performance comparison between Adam and Nesterov Momentum optimization methods. Comparing with Nesterov Momentum method, Adam performs similarly on CIFAR10 and SVHN, but worse on CIFAR100. Note that there are roughly 60000, 5000, and 500 training images for each class in SVHN, CIFAR10, and CIFAR100 respectively. This implies that Adam may be a better option for training CrescendoNet when the training data is abundant, due to the convenience of its adaptive learning rate scheduling.

The last row of Table 1 gives the result from path-wise training. Training the model with less memory requirement can be achieved at the cost of some performance degradation. However, Path-wise trained CrescendoNet still outperform many of networks without residual connections on given datasets.

### 3.4 RESULTS OF SUBNETS

Table 2 provides a performance comparison among different path combinations of CrescendoNet, trained by Adam optimization, with block-wise width $(128, 256, 512)$. The results show the ensemble behavior of our model. Specifically, the more paths contained in the network, the better the

Table 1: **Whole net classification error (%) on CIFAR10/CIFAR100/SVHN.** We highlight the top three accuracies in each column with the bold font. The three numbers in the parentheses denote the number of output feature maps of each block. The plus sign (+) denotes the data augmentation. The sign (-W) means that the feature maps of layers in each branch increase as explained in the model configuration section. The compared models include: Network in Network (Srivastava et al., 2015b), ALL-CNN (Springenberg et al., 2014), Deeply Supervised Net (Lee et al., 2014), Highway Network (Srivastava et al., 2015b), FractalNet (Larsson et al., 2017), ResNet (He et al., 2016a), ResNet with Stochastic Depth (Huang et al., 2016b), Wide ResNet (Zagoruyko & Komodakis, 2016), and DenseNet (Huang et al., 2016a).

| Method | Depth | Params | C10 | C10+ | C100 | C100+ | SVHN |
|---|---|---|---|---|---|---|---|
| Network in Network | - | - | 10.41 | 8.81 | 35.68 | - | 2.35 |
| All-CNN | - | - | 9.08 | 7.25 | - | 33.71 | - |
| Deeply Supervised Net | - | - | 9.69 | 7.97 | - | 34.57 | 1.92 |
| Highway Network | - | - | - | 7.72 | - | 32.39 | - |
| FractalNet (dropout+drop-path) | 21 | 38.6M | 7.33 | **4.60** | 28.20 | 23.73 | 1.87 |
| ResNet | 110 | 1.7M | 13.63 | 6.41 | 44.74 | 27.22 | 2.01 |
| Stochastic Depth | 110 | 1.7M | 11.66 | 5.23 | 37.80 | 24.58 | **1.75** |
| Wide ResNet | 16 | 11.0M | - | 4.81 | - | **22.07** | - |
| | 28 | 36.5M | - | **4.17** | - | **20.50** | - |
|    with Dropout | 16 | 2.7M | - | - | - | - | **1.64** |
| ResNet (pre-activation) | 164 | 1.7M | - | 5.46 | - | 24.33 | - |
| | 1001 | 10.2M | - | 4.62 | - | 22.71 | - |
| DenseNet (k = 12) | 40 | 1.0M | 7.00 | 5.24 | **27.55** | 24.42 | 1.79 |
| DenseNet-BC (k = 24) | 250 | 15.3M | **5.19** | **3.62** | **19.64** | **17.60** | 1.74 |
| CrescendoNet Nesterov | | | | | | | |
|   (128, 128, 128) | 15 | 4.1M | 7.26 | 5.53 | 29.83 | 25.09 | 1.90 |
|   (128, 256, 512)-W | 15 | 18.3M | 7.08 | 5.20 | 27.48 | 23.57 | 1.90 |
|   (128, 256, 512) | 15 | 27.7M | **6.81** | 5.03 | **26.39** | 22.97 | 1.78 |
|     without drop-path | 15 | 27.7M | 8.80 | 6.42 | 29.14 | 23.94 | 2.04 |
| CrescendoNet Adam | | | | | | | |
|   (128, 128, 128) | 15 | 4.1M | 7.26 | 5.20 | 33.04 | 25.76 | **1.73** |
|   (128, 256, 512) | 15 | 27.7M | **6.90** | 4.81 | 30.00 | 24.67 | 1.76 |
|     without drop-path | 15 | 27.7M | 9.20 | 6.90 | 33.50 | 26.35 | 1.79 |
|     path-wise training | 15 | 27.7M | 8.93 | 6.90 | 34.88 | 29.95 | 1.95 |

Table 2: **Subnet classification error (%) on CIFAR10/CIFAR100/SVHN.** The numbers in the curly brackets denote the branches used in each block.

| Branches per block | Depth | C10 | C10+ | C100 | C100+ | SVHN |
|---|---|---|---|---|---|---|
| P={1, 2, 3, 4} | 15 | 6.90 | 4.81 | 30.00 | 24.67 | 1.76 |
| P={2, 3, 4} | 15 | 6.91 | 4.93 | 29.90 | 24.92 | 1.87 |
| P={1, 2, 4} | 15 | 7.61 | 5.59 | 32.25 | 26.65 | 1.94 |
| P={1, 2, 3} | 12 | 7.94 | 6.00 | 31.86 | 27.18 | 2.02 |
| P={3, 4} | 15 | 7.54 | 5.31 | 31.61 | 26.29 | 1.97 |
| P={2, 4} | 15 | 7.73 | 5.56 | 32.60 | 27.09 | 2.01 |
| P={2, 3} | 12 | 8.03 | 5.85 | 32.08 | 28.24 | 2.04 |
| P={1, 4} | 15 | 8.66 | 6.38 | 35.81 | 29.74 | 2.05 |
| P={1, 2} | 9 | 10.58 | 8.69 | 37.03 | 34.08 | 2.75 |
| P={4} | 15 | 10.69 | 7.96 | 38.66 | 33.71 | 2.53 |
| P={3} | 12 | 11.31 | 8.27 | 38.26 | 34.70 | 2.43 |
| P={2} | 9 | 12.13 | 10.14 | 40.32 | 37.05 | 2.78 |
| P={1} | 6 | 28.60 | 30.31 | 70.51 | 73.41 | 8.74 |

performance. And the whole net outperforms any single path network with a large margin. For example, the whole net and the net based on the longest path show the inference error rate of 6.90% and 10.69% respectively, for CIFAR10 without data augmentation. This implicit ensemble behavior

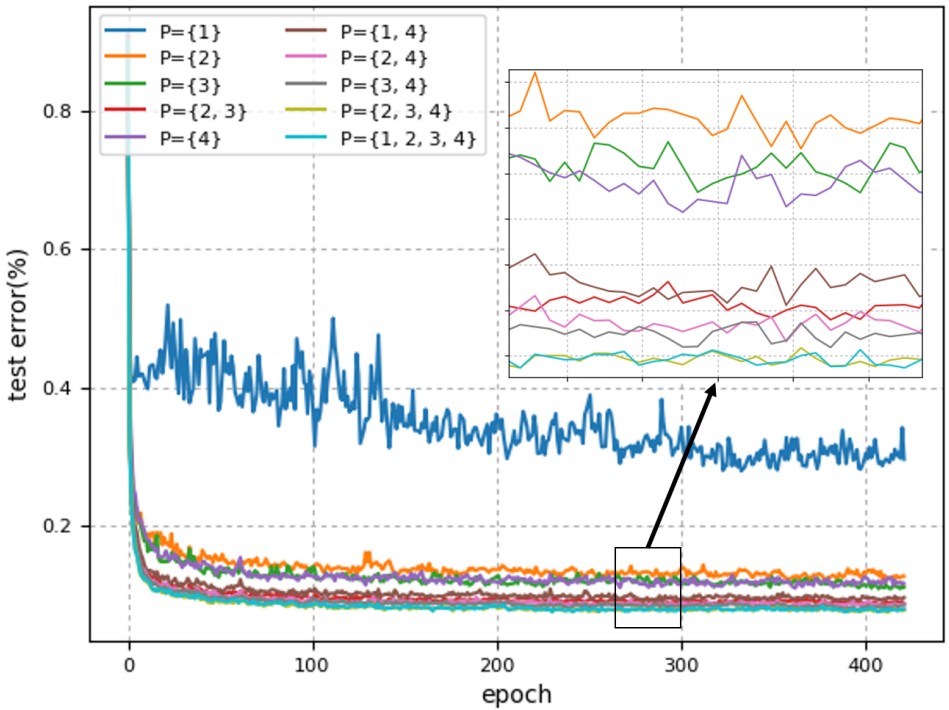

Figure 3: Error rates of subnets with different branch combinations when training with CIFAR10.

differentiates CrescendoNet from FractalNet, which shows a student-teacher effect. Specifically, the longest path in FractalNet can achieve a similar or even lower error rate compared to the whole net. To investigate the dynamic behavior of subnets, we test the error rate changes of subnets during the training. We use Adam to train the CrescendoNet with the structure shown in Figure 1 on CIFAR10 for 450 epochs. Figure 3 illustrates the behavior of different path combinations during the training. It shows that the inference accuracy of the whole net grows simultaneously with all the subnets, which demonstrates the ensemble effect. Second, for any single path network, the performance grows with the depth. This behavior of the anytime classifier is also shown by FractalNet. In other words, we could use the short path network to give a rough but quick inference, then use more paths to gradually increase the accuracy. This may be useful for time-critical applications, like integrated recognition system for autonomous driving tasks.

## 4 RELATED WORK

Conventional deep CNNs, such as AlexNet (Krizhevsky et al., 2012) and VGG-19 (Simonyan & Zisserman, 2014), directly stacked the convolutional layers. However, the vanishing gradient problem makes it difficult to train and tune very deep CNN of conventional structures. Recently, stacking small convolutional blocks has become an important method to build deep CNNs. Introducing new building blocks becomes the key to improve the performance of deep CNN. Lin et al. (2013) first introduced the NetworkInNetwork module which is a micro neural network using a multiple layer perceptron (MLP) for local modeling. Then, they piled the micro neural networks into a deep macro neural network.

Szegedy et al. (2015) introduced a new building block called Inception, based on which they built GoogLeNet. Each Inception block has four branches of shallow CNNs, building by convolutional kernels with size $1 \times 1$, $3 \times 3$, $5 \times 5$, and max-pooling with kernel size $3 \times 3$. Such a multiple-branch scheme is used to extract diversified features while reducing the need for tuning the convolutional sizes. The main body of GoogLeNet has 9 Inception blocks stacked each other. Stacking multiple-branch blocks can create an exponential combination of feed-forward paths. Such a structure com-

bined with the dropout technique can show an implicit ensemble effect (Veit et al., 2016; Srivastava et al., 2014). GoogLeNet was further improved with new blocks to more powerful models, such as Xception (Chollet, 2016) and Inception-v4 (Szegedy et al., 2016). To improve the scalability of GoogLeNet, Szegedy et al. (2016) used convolution factorization and label-smoothing regularization in Inception-v4. In addition, Chollet (2016) explicitly defined a depth-wise separable convolution module replacing Inception module.

Recently, Larsson et al. (2017) introduced FractalNet built by stacked Fractal blocks, which are the combination of identical convolutional layers in a fractal expansion fashion. FractalNet showed that it is possible to train very deep neural network through the network architecture design. FractalNet implicitly also achieved deep supervision and student-teacher learning by the fractal architecture. However, the fractal expansion form increases the number of convolution layers and associated parameters exponentially. For example, the original FractalNet model with 21 layers has 38.6 million parameters, while a ResNet of depth 1001 with similar accuracy has only 10.2 million parameters (Huang et al., 2016a). Thus, the exponential expansion reduced the scalability of FractalNet.

Another successful idea in network architecture design is the use of skip-connections (He et al., 2016a;b; Huang et al., 2016a; Zagoruyko & Komodakis, 2016; Xie et al., 2016). ResNet (He et al., 2016a) used the identity mapping to short connect stacked convolutional layers, which allows the data to pass from a layer to its subsequent layers. With the identity mapping, it is possible to train a 1000-layer convolutional neural network. Huang et al. (2016a) recently proposed DenseNet with extremely residual connections. They connected each layer in the Dense block to every subsequent layer. DenseNet achieved the best performance on benchmark datasets so far. On the other hand, Highway networks (Srivastava et al., 2015a) used skip-connections to adaptively infuse the input and output of traditional stacked neural network layers. Highway networks have helped to achieve high performance in language modeling and translation.

## 5 DISCUSSION AND CONCLUSION

CNN has shown excellent performance on image recognition tasks. However, it is still challenging to tune, modify, and design an CNN. We propose CrescendoNet, which has a simple convolutional neural network architecture without residual connections (He et al., 2016a). Crescendo block uses convolutional layers with same size $3 \times 3$ and joins feature maps from each branch by the averaging operation. The number of convolutional layers grows linearly in CrescendoNet while exponentially in FractalNet (Larsson et al., 2017). This leads to a significant reduction of computational complexity.

Even with much fewer layers and a simpler structure, CrescendoNet matches the performance of the original and most of the variants of ResNet on CIFAR10 and CIFAR100 classification tasks. Like FractalNet (Larsson et al., 2017), we use dropout and drop-path as regularization mechanisms, which can train CrescendoNet to be an anytime classifier, namely, CrescendoNet can perform inference with any combination of the branches according to the latency requirements. Our experiments also demonstrated that CrescendoNet synergized well with Adam optimization, especially when the training data is sufficient. In other words, we can avoid scheduling the learning rate which is usually performed empirically for training existing CNN architectures.

CrescendoNet shows a different behavior from FractalNet in experiments on CIFAR10/100 and SVHN. In FractalNet (Larsson et al., 2017), the longest path alone achieves the similar performance as the entire network, far better than other paths, which shows the student-teacher effect. The whole FractalNet except the longest path acts as a scaffold for the training and becomes dispensable later. On the other hand, CrescendoNet shows that the whole network significantly outperforms any set of it. This fact sheds the light on exploring the mechanism which can improve the performance of deep CNNs by increasing the number of paths.

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
