# OpenReview forum: "CrescendoNet: A Simple Deep Convolutional Neural Network with Ensemble Behavior"
_ICLR.cc/2018/Conference — Reject_

### Official Review · AnonReviewer1 · 2017-11-10
**Good network that is not sufficiently different from FractalNet and underperforms DiracNet**

**Rating:** 4
**Confidence:** 5

**Review:**

The paper presents a new CNN architecture: CrescendoNet. It does not have skip connections yet performs quite well.

Overall, I think the contributions of this paper are too marginal for acceptance in a top tier conference.

The architecture is competitive on SVHN and CIFAR 10 but not on CIFAR 100. The performance is not strong enough to warrant acceptance by itself.

FractalNets amd DiracNets (https://arxiv.org/pdf/1706.00388.pdf) have demonstrated that it is possible to train deep networks without skip connections and achieve high performance. While CrescendoNet seems to slightly outperform FractalNet in the experiments conducted, it is itself outperformed by DiracNet. Hence, CrescendoNet does not have the best performance among skip connection free networks.

You claim that FractalNet shows no ensemble behavior. This is clearly not true because FractalNet has ensembling directly built in, i.e. different paths in the network are explicitly averaged. If averaging paths leads to ensembling in CrescendoNet, it leads to ensembling in FractalNet. While the longest path in FractalNet is stronger than the other members of the ensemble, it is nevertheless an ensemble. Besides, as Veit showed, ResNet also shows ensemble behavior. Hence, using ensembling in deep networks is not a significant contribution.

The authors claim that "Through our analysis and experiments, we note that the implicit ensemble behavior of CrescendoNet leads to high performance". I don't think the experiments show that ensemble behavior leads to high performance. Just because a network performs averaging of different paths and individual paths perform worse than sets of paths doesn't imply that ensembling as a mechanism is in fact the cause of the performance of the entire architecture. Similary, you say "On the other hand, the ensemble model can explain the performance improvement easily." Veit et al only claimed that ensembling is a feature of ResNet, but they did not claim that this was the cause of the performance of ResNet.

Path-wise training is not original enough or indeed different enough from drop-path to count as a major contribution.

You claim that the number of layers "increase exponentially" in FractalNet. This is misleading. The number of layers increases exponentially in the number of paths, but not in the depth of the network. In fact, the number of layers is linear in the depth of the network. Since depth is the meaningful quantity here, CrescendoNet does not have an advantage over FractalNet in terms of layer number. Also, it is always possible to simply add more paths to FractalNet if desired without increasing depth. Instead of using 1 long paths, one can simply use 2, 3, 4 etc. While this is not explicitly mentioned in the FractalNet paper, it clearly would not break the design principle of FractalNet which is to train a path of multiple layers by ensembling it with a path of fewer layers. CrescendoNets do not extend beyond this design principle.

You say that "First, path-wise training procedure significantly reduces the memory requirements for convolutional layers, which constitutes the major memory cost for training CNNs. For example, the higher bound of the memory required can be reduced to about 40% for a Crescendo block with 4 paths where interval = 1." This is misleading, as you need to store the weights of all convolutional layers to compute the forward pass and the majority of the weights of all convolutional layers to compute the backward pass, no matter how many weights you intend to update. In a response to a question I posed, you mentioned that we you meant was "we use about 40% memory for the gradient computation and storage". Fair enough, but "gradient computation and storage" is not mentioned in the paper. Also, the reduction to 40% does not apply e.g. to vanilla SGD because the computed gradient can be immediately added to the weights and does not need to be stored or combined with e.g. a stored momentum term.

Finally, nowhere in the paper do you mention which nonlinearities you used or if you used any at all. In future revisions, this should be rectified.

While I can definitely imagine that your network architecture is well-designed and a good choice for image classification tasks, there is a very saturated market of papers proposing various architectures for CIFAR-10 and related datasets. To be accepted to ICLR, either outstanding performance or truly novel design principles are required.

---

> ### Author Response · Authors · 2018-01-05
> **CrescendoNet uses a different design principle from which of FractalNet and is better than it in terms of performance, architecture complexity, and model size**
>
> Dear reviewer,
>
> Thank you for your reviews.
>
> CrescendoNet did not only outperform FractalNet, but also has fewer parameters (27.7M vs. 36.5M), blocks (3 vs. 5), and shallower depth (15 vs. 21 layers). Crescendo architecture is simpler than Fractal architecture. The performance improvement makes CrescendoNet more competitive than FractalNet, and the linear growth of layers with respect to the number of branches significantly improves the model efficiency. Also, the performance behaviors of path combination are different between CrescendoNet and FractalNet. FractalNet shows a student-teacher effect while CrescendoNet does not. CrescendoNet may look like a variant of FractalNet, but its design principal and network properties are different.
>
> CrescendoNet outperforms deep networks without residual connections on CIFAR10, CIFA100, and SVHN without data augmentation in given experiments. Although DiracNet slightly outperforms CrescendoNet on CIFAR with data augmentation, it has about twice numbers of layers (34 vs. 15) and parameters (59M vs. 27M) as CrescendoNet does.
>
> You are right. Our original claim that "CrescendoNet shows ensemble behavior while FractalNet shows student-teacher effect" is misleading. Thus we have removed or modified relevant parts in revision such as: "On the other hand, CrescendoNet shows that the whole network significantly outperforms any set of it. This fact sheds light on exploring the mechanism which can improve the performance of deep CNNs by increasing the number of paths." We discovered such a difference and thought it is worth further studying.
>
> Thank you for pointing out that our claim that "Through our analysis and experiments, we note that the implicit ensemble behavior of CrescendoNet leads to high performance" is not solid. We have replaced it with the following statement in revision. "Through our analysis and experiments, we discovered an emergent behavior which is significantly different from which of FractalNet. The entire CrescendoNet outperforms any subset of it can provide an insight of improving the model performance by increasing the number of paths by a pattern."
>
> We think you made some assumptions for FractalNet in the following statements.
>
> "Also, it is always possible to simply add more paths to FractalNet if desired without increasing depth. Instead of using 1 long paths, one can simply use 2, 3, 4 etc."
>
> For the above statement, we agree that it is always possible to do this. However, this will not only break the fractal expansion pattern by involving manually designed parts but also cannot guarantee the good performance.
>
> "While this is not explicitly mentioned in the FractalNet paper, it clearly would not break the design principle of FractalNet which is to train a path of multiple layers by ensembling it with a path of fewer layers. CrescendoNets do not extend beyond this design principle."
>
> We have a different understanding of the design principle of FractalNet. The author of FractalNet describes his design principle in Abstract: "We introduce a design strategy for neural network macro-architecture based on self similarity. Repeated application of a simple expansion rule generates deep networks whose structural layouts are precisely truncated fractals." It is clear that the design principle of FractalNet is the fractal expansion rule instead of "to train a path of multiple layers by ensembling it with a path of fewer layers" which applies to our model more. The recursive model formula and model illustration also demonstrate the design principle of FractalNet. However, we use the linear expansion rule, which is different from the Fractal design principle. Also, if the design principle can apparently lead to the design of CrescendoNet, the author of FractalNet would like to propose it since it is cleaner and better-performed.
>
> Thank you very much for pointing out our misleading statement about path-wise training. We have corrected relevant parts by explicitly saying we can only save computation and storage memory for convolution layers when using momentum optimizers.
>
> Also, we appreciate you for asking about the missing architecture and experiment design details. We have added the detailed description of fully connected layers and nonlinearities.
>
> Thank you again for your careful and professional reviews. Your questions and comments definitely helped us to improve the work.

---

> > ### Comment · AnonReviewer1 · 2018-01-08
> > **Not convinced**
> >
> > You state that you have higher accuracy than DenseNet-40, lower depth and DenseNet-250, higher accuracy than FractalNet, lower depth than DiracNet etc. However, it is not enough to argue that CrescendoNet is not strictly worse than other architectures in order to argue that CrescendoNet should be used in practice. I don't think the results in the paper support such a case.
> >
> > I am not convinced that the behavior / design / scientific meaning of CrescendoNet is sufficiently novel compared to FractalNet / DiracNet.

---

### Official Review · AnonReviewer2 · 2017-11-27

**Rating:** 5
**Confidence:** 4

**Review:**

This paper proposes a new convolutional network architecture, which is tested on three image classification tasks.

Pros:
The network is very clean and easy to implement, and the results are OK.

Cons:
The idea is rather incremental compared to FractalNet. The results seem to be worse than existing networks, e.g., DenseNet (Note that SVHN is no longer a good benchmark dataset for evaluating state-of-the-art CNNs). Not much insights were given.

One additional question: Skip connections have been shown to be very useful in ConvNets. Why not adopt it in CrescendoNet? What's the point of designing a network without skip connections?

---

> ### Author Response · Authors · 2018-01-05
> **Responses to questions**
>
> Dear reviewer,
>
> Thank you for your reviews and suggestions.
>
> Although CrescendoNet with 15 layers performed worse than DenseNet-BC with 250 layers, it outperformed DenseNet with 40 layers using all given datasets.
>
> Thank you for the suggestion of using skip-connections. We are working on applying skip-connection as a module to our architecture. The challenge is leveraging the skip-connection to improve the performance while keeping the architecture clean and efficient.
>
> As for the motivation, designing a network without skip-connections is our first step. In other words, we first proposed a design pattern with a base model and then we keep trying to incorporate existing and new techniques, such as Residual connections, bottleneck layers, and depth-wise separable convolutions.
>
> Thank you again for your reviews and suggestions.

---

### Official Review · AnonReviewer3 · 2017-11-27
**The authors propose  a new network architecture without residual connections. The motivation of the paper is not clear. The contribution of the paper is not sufficient.**

**Rating:** 4
**Confidence:** 5

**Review:**

In this paper, the authors propose  a new network architecture, CrescendoNet, which is a simple stack of building blocks without residual connections. To reduce the memory required for training, the authors also propose a path-wise training procedure based on the independent convolution paths of CrescendoNet. The experimental results on CIFAR-10, CIFAR-100 and SVHN show that CrescendoNet outperforms most of the networks without residual connections.

Contributions:

1 The authors proposed Crescendo block that consists of convolution paths with increasing depth.

3 The authors conducted experiments on three benchmark datasets and show promising performance of CrescendoNet .

3 The authors proposed a path-wise training procedure to reduce memory requirement in training.

Negative points:

1 The motivation of the paper is not clear. It is well known that the residual connections are important in training deep CNNs and have shown remarkable performance on many tasks. The authors propose the CrescendoNet which is without residual connections. However, the experiments show that CrescendoNet is worse than ResNet.

2  The contribution of this paper is not clear. In fact, the performance of CrescendoNet is worse than most of the variants of residual networks, e.g., Wide ResNet, DenseNet, and ResNet with pre-activation. Besides, it seems that the proposed path-wise training procedure also leads to significant performance degradation.

3 The novelty of this paper is insufficient. The CrescendoNet is like a variant of the FractalNet, and the only difference is that the number of convolutional layers in Crescendo blocks grows linearly.

4 The experimental settings are unfair. The authors run 700 epochs and even 1400 epochs with path-wise training on CIFAR, while the baselines only have 160~400 epochs for training.

5 The authors should provide the experimental results on large-scale data sets (e.g. ImageNet) to prove the effectiveness of the proposed method, as they only conduct experiments on small data sets, including CIFAR-10, CIFAR-100, and SVHN.


6 The model size of CrescendoNet is larger than residual networks with similar performance.


Minor issues:

1In line 2, section 3.4, the period after “(128, 256, 512)” should be removed.

---

> ### Author Response · Authors · 2018-01-05
> **Clarification of motivation, contribution, and experiment details**
>
> Dear reviewer,
>
> We appreciate your time and review.
>
> The motivation of our CrescendoNet is designing a CNN architecture that achieves rich feature representation and is easy to implement. We designed the model without skip-connections as our first step.  In the future, we can continue to improve the model by incorporating existing and new techniques, such as Residual connections, bottleneck layers, and depth-wise separable convolutions.
>
> Regarding the performance, CrescendoNet outperforms most of the models without residual connections. Comparing with ResNet and its variants, the performances of CrescendoNet with 15 layers is better than those of original ResNet with 110 layers, ResNet(pre-activation) with 164 layers, DenseNet with 40 layers and matches Wide ResNet with 16 layers. Only ResNet variants with extreme depth ranging from 110 to 1001 outperform CrescendoNet Since the architectures of these ResNet variants are manually designed, the deeper the model is, the less usability, modifiability, and extensibility can be. In contrast, a clean and efficient architecture, e.g., CrescendoNet, has advantages in practical applications. For example, the state-of-the-art object detection model, YOLO-V2, still adopts a VGGNet-like CNN architecture called DarkNet instead of ResNet variants.
>
> Comparing with FractalNet, CrescendoNet has fewer parameters, blocks, and shallower depth but better performance. The performance improvement makes CrescendoNet more competitive than FractalNet, and the linear growth of layers significantly improves the model efficiency. CrescendoNet may look like a variant of FractalNet, but its design principal and network properties are different.
>
> As for the number of training epochs, we think it is more relevant to the learning rate schedule and optimizers used. But you are right. It is our responsibility to reduce the training time as much as possible.
>
> We have corrected the typo you pointed out.
>
> Thank you again for your reviews and corrections.

---

### Comment · AnonReviewer1 · 2017-11-10
**Reviewer question**

Dear authors,

I would like to confirm you is that you do not use any ReLU or other nonlinearities at all in the convolutional part of the network. Also, how many fully-connected layers do you use and what, if any, nonlinearities do you use before, between or after them?

Also, how did you initialize the weights in each layer?

Thanks,

---

> ### Author Response · Authors · 2017-11-10
> **Dear reviewers,**
>
> We are very sorry for missing the description.
>
> For the activation function, the listed experiment results used ReLU as the activation function following each convolutional layer and fully connected layer. In addition, we also did some tests with the exponential linear unit (ELU) as the activation function, which turned out the performance is almost unchanged.
>
> We used two fully connected layers, with 384 hidden units and 192 hidden units respectively.
>
> For the weight initialization, we mentioned in the paper, in the first paragraph of Section 3.2 "We use truncated normal distribution for parameter initialization. The standard deviation of hyper-parameters is 0.05 for convolutional weights and 0.04 for fully connected layer weights."
>
> Thank you for pointing out the problem.

---

### Comment · AnonReviewer1 · 2017-11-10
**Another reviewer question**

Dear authors,

you say that path-wise training can reduce memory usage by 40%. How is this? If I freeze the weights in all but one path, I still need to have those weights in memory to compute the forward and backward pass of the network. Just because the weights in some layers aren't updated doesn't mean they aren't needed in memory to compute the activations and gradients of other weights.

Thanks,

---

> ### Author Response · Authors · 2017-11-10
> **About path-wise training and memory efficiency**
>
> Dear Reviewers,
>
> Thank you for the reply.
>
> You are definitely right. The loss is computed with all the convolutional layers involved. However, we didn't state that the memory saving is because we don't need to keep all the parameters in memory for the forward and backward pass. We may cause this misunderstanding by the unclear description. Here, we clarify that we save the memory by avoiding the memory for computing and keeping the gradients for the frozen layers.
>
> For the whole net training using the back propagation, we compute gradients for all the convolutional kernels and keep them in the memory. Then we update all the parameters with corresponding gradients.
>
> For the path-wise training, we only need to compute the gradients for a portion of convolutional kernels. We do use the frozen parameters, but we avoid the memory used for computing and storing the gradients for the frozen parameters (note that one parameter needs one gradient).
>
> In section 2.2, we gave an example that the memory requirement for training convolutional layers can be reduced to 40%, when training Crescendo blocks with four branches of lengths: (1, 2, 3, 4). For whole net training, we need to compute and keep ((1+2+3+4) * number_of_feature_maps_per_layer * kernel_size) gradients in each backward pass, while we only need ((4) * number_of_feature_maps_per_layer * kernel_size) gradients when training the deepest path by path-wise training. In this sense, we use about 40% memory for the gradient computation and storage during the back propagation. We are sorry for the misunderstanding.
>
> Thank you for pointing out our problems.

---

### Comment · AnonReviewer1 · 2017-11-10
**Can you see the review?**

Dear authors,

Thank you for answering my 2 questions. I just posted my review. I am curious: Can you see the review? Because when I log out of my account, I can no longer see it. Hence, the review isn't public. I am wondering whether at least you can see it.

Thanks,

---

> ### Author Response · Authors · 2017-11-10
> **We can see your review**
>
> Dear reviewers,
>
> We can see your review. We highly appreciate your help for improving our work.
>
> Thank you for reviewing our paper.

---

### Public Comment · (anonymous) · 2017-11-22
**Compare the CrescendoNet architecture with deeply-fused nets**

Dear authors,

Can you compare the CrescendoNet architecture with deeply-fused nets, https://arxiv.org/abs/1605.07716?

Thanks,

---

> ### Author Response · Authors · 2017-11-23
> **CrescendoNet VS Deeply fused nets**
>
> Dear reader,
>
> Thank you for the great question!
>
> There are three main differences between CrescendoNet and Deeply fused net (DFN):
>
>     1. design manually vs. design by patterns
>     DFN blocks have manually defined structures, which are different from each other. CrescendoNet proposes a design pattern, which generates the block architecture by two hyper-parameters and all the blocks have the same structure.
>
>     The base model of DFN is a combination of 7 networks with different numbers of layers (19, 50, 5, 8, 10, 11, 14). DFN segments each of 7 networks into three parts and fuses the corresponding segments into blocks. Also, DFN applies different fusion schemes for its blocks. For example, the first block contains branches with numbers of layers (5, 16, 1, 2, 2, 3, 4) while the second block has (6, 16, 1, 2, 3, 3, 4) layers for its branches. For the CrescendoNet base model, each block includes network branches of increasing number of layers (1, 2, 3, 4...), which defined by Scale and Interval. And the structure of each block is identical.
>
>     2. fusion vs. expansion
>     The central idea of DFN is combining pre-defined and separated single-path networks by fusing their intermediate feature representations. Thus, DFN uses one fully connected layer for each branch, which means DFN has seven individual fully connected layers for its base model. CrescendoNet generates the whole architecture by the expansion. CrescendoNet has two sequent fully connected layers following the body of the whole net.
>
>     Although both fusion and expansion may achieve ensemble behavior, the fusion process in DFN is a generalization of ResNet and Highway net, while CrescendoNet may achieve the ensemble and feature diversification through designed structure patterns.
>
>     3. performance comparison
>     For CIFAR10 and CIFAR100 datasets with widely-used data augmentation scheme, CrescendoNet with 15 layers outperforms DFN with 50 layers by a large margin. The best error rates of DFN and CrescendoNet on CIFAR10 are 6.02% and 4.81% respectively and on CIFAR100 are 27.36% and 22.97%.
>
>     The Figure 1 in the DFN paper may cause readers think CrescendoNet has the same structure with DFN. However, the figure is an illustration of the deep (may mean "on intermediate layers") fusion from different networks. There are no explicit design rules given to specify the architecture of the whole net. I think that is because pre-defined independent networks partially determine the architecture of DFN.
>
>     We summarize some small differences between two architectures as follows:
>     •  DFN uses the same initialization scheme as VGG net used, while CrescendoNet uses the truncated normal distribution.
>     •  DFN uses average pooling while CrescendoNet not.
>     •  DFN uses size 2x2 max pooling while Crescendo uses 3x3 (but this doesn't matter observed from our experiment results).
>
>
> Wang, Jingdong, Zhen Wei, Ting Zhang, and Wenjun Zeng. "Deeply-fused nets." arXiv preprint arXiv:1605.07716 (2016).

---

> > ### Public Comment · (anonymous) · 2017-11-24
> > **CrescendoNet VS Deeply fused nets**
> >
> >  1. design manually vs. design by patterns
> >  DFN presents a framework using the deep fusion concept. It is also a design pattern. Figure 1(b) in DFN uses a simple design pattern: each block is the same, not different.
> >
> > From the DFN paper, I did not find the description about the following comment by the author:
> > "The base model of DFN is a combination of 7 networks with different numbers of layers (19, 50, 5, 8, 10, 11, 14). DFN segments each of 7 networks into three parts and fuses the corresponding segments into blocks. Also, DFN applies different fusion schemes for its blocks. For example, the first block contains branches with numbers of layers (5, 16, 1, 2, 2, 3, 4) while the second block has (6, 16, 1, 2, 3, 3, 4) layers for its branches."
> >
> > 2. fusion vs. expansion
> > I think this is just a difference in terms, but the essential is almost the same. The fusion process in DFN is not simply a generalization of ResNet and Highway net, but it provides the possible reason why this kind of network structures could be trained easily.
> >
> > About diversity, I found that the following paper provides interesting analysis:
> > On the Connection of Deep Fusion to Ensembling. Liming Zhao, Jingdong Wang, Xi Li, Zhuowen Tu, Wenjun Zeng. https://arxiv.org/abs/1611.07718v1

---

> > > ### Author Response · Authors · 2017-11-24
> > > **CrescendoNet VS Deeply fused nets**
> > >
> > > Dear reader,
> > >
> > > Thank you for the comments.
> > >
> > > 1. design manually vs. design by patterns
> > > Admittedly, everyone can have his/her definition of the concept of the design pattern. However, to use deeply fusion method, we need manually design the architectures, the segmentation positions and fusion schemes of individual networks.
> > >
> > > In addition, Figure 1(b) is an illustration of the deep fusion concepts in contrast with the shallow fusion. There is no description that the block structures are identical at all. Also, all the architecture examples given in the paper don’t have all blocks with the same structure, according to Table 1 and 2.
> > >
> > > For the statement:
> > > "The base model of DFN is a combination of 7 networks with different numbers of layers (19, 50, 5, 8, 10, 11, 14). DFN segments each of 7 networks into three parts and fuses the corresponding segments into blocks. Also, DFN applies different fusion schemes for its blocks. For example, the first block contains branches with numbers of layers (5, 16, 1, 2, 2, 3, 4) while the second block has (6, 16, 1, 2, 3, 3, 4) layers for its branches."
> > > We can get the above statements from Table 1.
> > > Caption: “Table 1. Base network architectures. “
> > > Row #2: “#Layers: 19, 50, 5, 8, 10, 11, 14” This means the fused nets have corresponding numbers of layers.
> > > Row #4: “C1. ... 5, 16, 1, 2, 2, 3, 4” This means the segments/branches have corresponding layers.
> > > Row #6: “c2. ... 6, ...” Similarly.
> > >
> > > 2. fusion vs. expansion
> > > We agree that fusion is a helpful way for training.
> > >
> > > Thank you very much for paying attention to our paper and providing the interesting paper. Please feel free to discuss with us if you have any other ideas.

---

### Decision · Program_Chairs · 2018-01-29
**ICLR 2018 Conference Acceptance Decision**

**Decision:**

Reject

**Comment:**

The paper proposes a new convolutional network architecture, called CrescendoNet. Whilst achieving competitive performance on CIFAR-10 and SVHN, the accuracy of the proposed model on CIFAR-100 is substantially lower than that of state-of-the-art models with fewer parameters; the paper presents no experimental results on ImageNet. The proposed architecture does not provide clear new insights or successful new design principles. This makes it unlikely the current manuscript will have a lot of impact.